# Unraveling sulfur chemistry in interstellar carbon oxide ices

Xiaolong Li [1], Bo Lu[1], Lina Wang[1], Junfei Xue[1], Bifeng Zhu[1], Tarek Trabelsi[2], Joseph S. Francisco [2] ✉ & Xiaoqing Zeng [1] ✉

Formyl radical (HCO•) and hydroxycarbonyl radical (HOCO•) are versatile building blocks in the formation of biorelevant complex organic molecules (COMs) in interstellar medium. Understanding the chemical pathways for the formation of HCO• and HOCO• starting with primordial substances (e.g., CO and $CO_2$) is of vital importance in building the complex network of prebiotic chemistry. Here, we report the efficient formation of HCO• and HOCO• in the photochemistry of hydroxidooxidosulfur radical (HOSO•)–a key intermediate in $SO_2$ photochemistry–in interstellar analogous ices of CO and $CO_2$ at 16 K through hydrogen atom transfer (HAT) reactions. Specifically, 266 nm laser photolysis of HOSO• embedded in solid CO ice yields the elusive hydrogen-bonded complexes HCO•···$SO_2$ and HOCO•···SO, and the latter undergoes subsequent HAT to furnish $CO_2$···HOS• under the irradiation conditions. Similar photo-induced HAT of HOSO• in solid $CO_2$ ice leads to the formation of HOCO•···$SO_2$. The HAT reactions of HOSO• in astronomical CO and $CO_2$ ices by forming reactive acyl radicals may contribute to understanding the interplay between the sulfur and carbon ice-grain chemistry in cold molecular clouds and also in the planetary atmospheric chemistry.

Sulfur is the tenth most abundant element in the universe, and sulfur chemistry plays vital importance not only in the biological systems and atmosphere on the Earth but also in interstellar medium (ISM). According to astronomical observations, the two sulfur oxides $SO_2$ and SO have been found to be abundant in molecular clouds[1–4], which are mainly formed through condensation of gas-phase molecules at the surface of dust grains (mostly amorphous silicates) with an onion-like structure at the temperatures of about 10–20 K[5,6]. The inner layer of the grains mainly consists of hydrogenated ice ($H_2O$) with low concentrations of other H-containing species such as $CH_3OH$, $NH_3$, and $CH_4$. The outer layer is made up of dehydrogenated ices with dominant compositions of CO, $CO_2$, $N_2$, $O_2$, and $SO_2$, and low concentrations of $H_2O$ may also be present in the outer layer of the icy mantle. The icy mantle at the surface of cosmic dust grains are the most important carriers of prebiotic molecules, and the composition of the mantles are largely affected by the exchanges between solid ice and gas-phase and also the photochemistry promoted by cosmic irradiations, including

UV and X-ray photons from nearby stars. Therefore, the study about the chemical composition of the icy grains and the complex reaction networks is crucial for understanding the evolution of the molecular clouds[7–9].

Carbon monoxide (CO) is the most abundant composition in the outer layer of icy grains in interstellar medium, and CO-abundant ices have also been found at the surface of many cold interstellar bodies, including comets, icy moons, and planets in the outer solar system. Therefore, the chemistry of CO through successive hydrogen atom addition reactions in the CO-rich outer layer of the interstellar icy grains may play a key role for the formation of complex organic molecules (COMs), which are probably building blocks for the origin of life[10–12]. Recently, it has been shown that simple radicals bearing elements C, N, O, P, or S play vital importance in prebiotic synthesis[13]. Reaction networks of these radicals in interstellar ice grains and the corresponding geochemical scenarios may help in unveiling the chemical evolution and origins of life. For instance, formyl radical (HCO•)

[1]Department of Chemistry, Shanghai Key Laboratory of Molecular Catalysts and Innovative Materials, Fudan University, 200433 Shanghai, China. [2]Department of Earth and Environment Science, University of Pennsylvania, Philadelphia, PA 19104-6243, USA. ✉e-mail: frjoseph@sas.upenn.edu; xqzeng@fudan.edu.cn

and hydroxycarbonyl radical (HOCO•) are important intermediates in atmospheric and combustion chemistry[14], and they are also versatile building blocks in the interstellar formation of biorelevant COMs such as formic acid (HC(O)OH)[15], glyoxylic acid (HC(O)C(O)OH)[16], and pyruvic acid (CH$_3$C(O)C(O)OH)[17] in low temperature (<30 K) interstellar CO and CO$_2$ ices doped with H-containing species CH$_4$ and H$_2$O through barrierless radical-radical association reactions, in which the reactive acyl radicals can be generated through the hydrogenation of carbon oxides with the H-containing molecules embedded in the same ice layer at cosmic radiations[18,19]. As a simple organic species, HCO• has been observed in many interstellar clouds such as DR 21, Sgr B2, and NGC 0024[20,21], and its radical recombination reaction with •CH$_2$OH in producing COMs during the phase transition of interstellar CO ices doped with CH$_3$OH and H$_2$O at a typical dense cloud temperature of about 10 K has been recently disclosed[22–24]. The cationic form of HOCO• has been also identified in star-forming regions such as SgrB2(OH) and low-mass protostar IRAS 04368 + 2557 in L1527[25,26].

In sharp contrast to the extensively explored mechanisms for the formation of COMs through the photoreactions of H-containing species (e.g., CH$_3$OH, NH$_3$, and CH$_4$) via the intermediacy of organic radicals such as HCO•, HOCO•, and CH$_3$O• in interstellar icy grain mantles, the ice-grain photochemistry of the typical dehydrogenated molecules such as SO, SO$_2$, and the derived sulfur-containing radicals HOS• and HOSO• in astronomical CO and CO$_2$ ices remains barely investigated. On the other hand, the photochemistry of SO and SO$_2$ also has great impact on the sulfur cycle in planetary atmospheres due to the formation and evolution of hazes and clouds in the upper atmospheres of Solar system planets such as Earth[27], Venus[28], Jupiter[29], and the moon Io[30]. Among these sulfur oxides, SO$_2$ is one of the most common pollutant in the Earth's atmosphere. Therefore, the SO$_2$ photochemistry has been the focus of enormous attention due to the important role in sulfur cycle by forming sulfuric acid and sulfate aerosols. According to the recent modeling studies[31,32], the tropospheric photochemistry of SO$_2$ in the presence of water proceeds mainly through the formation of HOSO• and hydroxyl radical (•OH) after absorption of light in the near UV–vis range (250–340 nm). In contrast, the UV photolysis (190–220 nm) of SO$_2$ in the gas phase leads to fragmentation by yielding SO[33], which is an interstellar species that has been detected in the atmospheres of Venus[34] and Io[35]. Chemically, SO is more reactive and it dimerizes easily to yield elemental sulfur and SO$_2$ via the intermediacy of OSSO, and the dimer has been recognized as a candidate species that contributes to the mysterious near-UV absorption (320–400 nm) in the yellowish atmosphere of Venus[36–38].

Herein, we report an experimental study on the photochemistry of the astrochemically relevant sulfur-containing species HOSO•, HOS•, SO$_2$, SO, and OSSO in solid CO and CO$_2$ ices at 16 K (Fig. 1). In addition to the molecular complexes formed between HOSO• and the carbon oxides, the photo-induced hydrogen atom transfer (HAT) to form new complexes consisting of acyl radicals (HCO• and HOCO•) and sulfur oxides (SO$_2$ and SO) has been observed. It is noteworthy that weakly bonded molecular complexes consisting of interstellar species have been considered as potent contributors to the rich chemistry in low-temperature giant molecular clouds[39].

## Results and Discussion

### Isolation of HOSO• in CO and CO$_2$ ices

Thanks to the strong "cage effect" of the solid host matrix materials (e.g., Ne and Ar) at low temperatures (<30 K), the matrix isolation technique has been broadly applied in trapping highly unstable intermediates, including weakly bonded molecule-radical complexes such as •OH···CO[18], •C$_6$H$_5$···H$_2$O[40], •OC$_6$H$_5$···H$_2$O[41], •OH···H$_2$O[42], and •NH$_2$···H$_2$O[43]. Recently, it has been shown that HOSO• can be efficiently generated in the gas phase through high-vacuum flash pyrolysis (HVFP, ca. 700 °C) of CHF$_2$S(O)OH (Fig. 2a), and photolysis of HOSO• (Fig. 2b) in solid Ar-matrix at 10 K yields isomeric HSO$_2$•, fragments SO$_2$/H•

together with the caged radical complex •OH···OS[44]. The absence of free fragments •OH and OS in the matrix suggests that they can hardly escape from the rigid matrix cages. When the pyrolytic generation of HOSO• was performed in the presence of CO by using a 1:20:1000 mixture of CHF$_2$S(O)OH/CO/Ar, the IR spectrum of the isolated species (Supplementary Fig. 1) shows the appearance of new IR bands in the range of 3500–3350 cm$^{-1}$ for O–H stretching vibrations ($v$(OH)), implying complex formation between HOSO• and CO.

When using neat CO as the matrix host material, the IR spectrum of the isolated pyrolysis products at 16 K (Fig. 2c) clearly shows the absence of all the IR bands for free HOSO• (Fig. 2b), while the bands for the complex CO···HOSO• become dominant. The $v$(OH) mode in HOSO• shifts from 3545.3 cm$^{-1}$ in Ar-matrix to 3396.8 cm$^{-1}$ in CO-matrix, corresponding to a red-shift ($\Delta v$) of −148.5 cm$^{-1}$. It is comparable with the shift of the $v$(OH) mode in CO-matrix isolated HOCO• ($\Delta v$ = −146.9 cm$^{-1}$) comparing to its IR spectrum in Ar-matrix[45]. The assignment of CO···HOSO• is supported by the good agreement with the theoretically calculated shift of −140 cm$^{-1}$ at the B3LYP-GD3(BJ)/def2-TZVP level (Table 1). In line with a stable hydrogen-bonded structure through OC···H bond in the complex, the stretching mode for the terminal S = O moiety ($v$(S = O)) is less perturbed than the S–O stretching mode ($v$(S–O)) as indicated by the shifts of −3.6 and +25.3 cm$^{-1}$, respectively. In contrast, the weak deformation mode $\delta$(SOH) exhibits a large blue-shift of +44.2 cm$^{-1}$, which is in agreement with the calculated shift of +53 cm$^{-1}$ at the CCSD(T)/aug-cc-pV(T + D)Z level. Note there are several weak satellite bands around the main peaks for the fundamental modes of the complex, they probably arise from the less stable matrix sites or the less abundant complexes of HOSO• consisting two or more CO molecules.

By analogy, deposition of the pyrolysis products of a 1:50:1000 mixture of CHF$_2$S(O)OH/CO$_2$/Ar at 16 K leads to the formation of the complex CO$_2$···HOSO• (Supplementary Fig. 2). Consistent with the CCSD(T) calculated red-shift of −73 cm$^{-1}$ (B3LYP: −84 cm$^{-1}$) for the $v$(OH) mode in the complex, the band for HOSO• shifts from 3545.3 cm$^{-1}$ in Ar-matrix to 3478.7 cm$^{-1}$ in CO$_2$-matrix (Table 1), corresponding to a shift of −66.6 cm$^{-1}$. Concomitantly, the $v$(S–O) mode undergoes a blue-shift by +22.4 cm$^{-1}$ (CCSD(T): + 21 cm$^{-1}$; B3LYP: + 24 cm$^{-1}$).

Weak hydrogen-bonding interactions of HOSO• with carbon oxides also affect its UV-vis absorption. Recently, a broad absorption

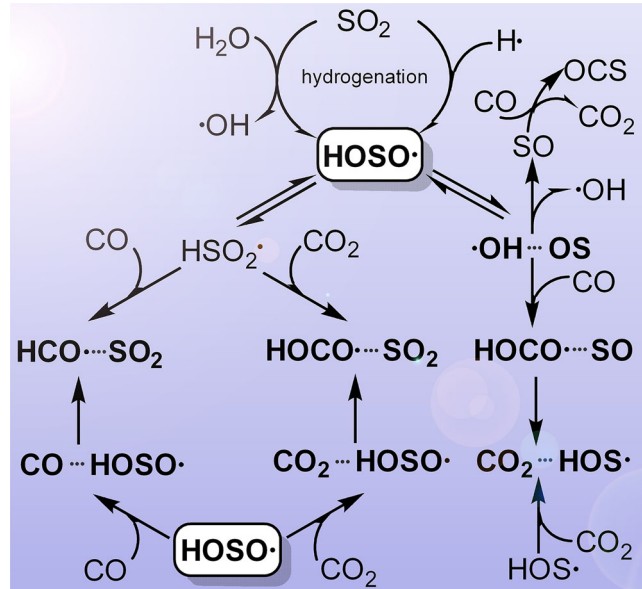

**Fig. 1 | Photochemistry of HOSO• in solid CO and CO$_2$ ices.** Proposed pathways for the photo-induced decomposition, isomerization, and hydrogen atom transfer reactions of HOSO• in solid CO and CO$_2$ at 16 K.

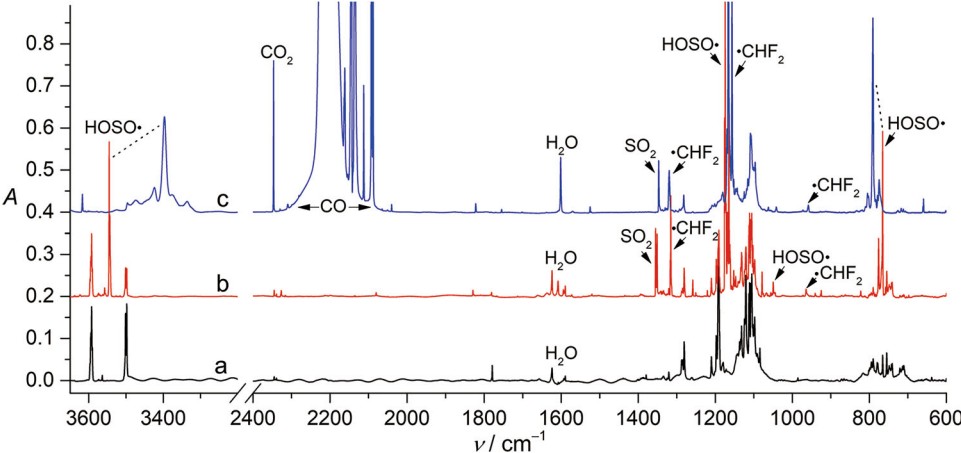

**Fig. 2 | Infrared (IR) spectrum of CHF₂S(O)OH and HOSO• in matrixes. a** IR spectrum of CHF₂S(O)OH in Ar-matrix at 10 K. **b** IR spectrum for the high-vacuum flash pyrolysis (HVFP, ca. 700 °C) products of CHF₂S(O)OH in Ar-matrix at 10 K. **c** IR spectrum for the HVFP (ca. 700 °C) products of CHF₂S(O)OH in CO-matrix at 16 K.

**Table 1 | Calculated and observed Infrared (IR) spectra of HOSO• in different matrixes**

| HOSO• | | | | CO···HOSO• | | | CO₂···HOSO• | | |
|---|---|---|---|---|---|---|---|---|---|
| Mode | Obs.[a] | B3LYP[b] | CCSD(T)[c] | Obs.[d] | B3LYP[b] | CCSD(T)[c] | Obs.[e] | B3LYP[b] | CCSD(T)[c] |
| $\nu$(O–H) | 3545.3 | 3717 (91) | 3736 | 3396.8 | 3577 (469) | 3640 | 3478.7 | 3633 (311) | 3663 |
| $\nu$(S = O) | 1168.2 | 1178 (104) | 1183 | 1164.6 | 1182 (103) | 1192 | 1163.8 | 1176 (136) | 1186 |
| $\delta$(SOH) | 1049.7 | 1062 (16) | 1081 | 1093.9 | 1125 (8) | 1134 | n.o. | 1108 (6) | 1121 |
| $\nu$(S–O) | 765.6 | 766 (188) | 781 | 790.9 | 788 (173) | 797 | 788.0 | 790 (167) | 802 |

[a] Observed IR frequencies in Ar-matrix. [b] Calculated harmonic IR frequencies and intensities (km mol⁻¹, in parentheses) at the B3LYP-GD3(BJ)/def2-TZVP level of theory. [c] Calculated harmonic IR frequencies at the CCSD(T)/aug-cc-pV(T + d)Z level of theory. [d] Observed IR frequencies in CO-matrix. [e] Observed IR frequencies in CO₂-doped Ar-matrix.

centered at 270 nm ($\lambda_{max}$) has been observed for HOSO• in Ar-matrix, corresponding to the transition from the ground state (X²A) to the C²A/D²A excited states[46]. As shown in Fig. 3, the major absorption of HOSO• in CO-matrix appears in the range of 350–240 nm as a weak band, and its assignment is ascertained with the photochemistry that HOSO• can be efficiently depleted by UV-light irradiation at 266 nm[47]. This characteristic absorption for HOSO• is also observable in CO₂-matrix. In sharp contrast to the appearance of the absorption of HSO₂• in the range of 320 – 500 nm after the irradiation (266 nm) of HOSO• in Ar-matrix, same photolysis in CO- and CO₂-matrixes results in the occurrence of weaker absorptions in the range of 300 – 400 nm, implying the formation of new species arising from the photoreactions of HOSO• with the carbon oxides.

**Photochemistry of HOSO• in CO and CO₂ ices**

To unravel the photochemistry of HOSO• in CO- and CO₂-matrixes, the IR spectra for the 266 nm laser photolysis products were recorded and the resulting difference spectra reflecting the reactions of HOSO• are depicted in Fig. 4. In contrast to the photodissociation of HOSO• to H•/SO₂ and the caged complex •OH···SO in Ar-matrix (Fig. 4a), its photolysis in solid CO (Fig. 4b) yields CO₂, OCS, HCO•, HOCO•, H₂CO, HOS•, and SO₂. It is noteworthy that the IR frequencies for all these species shift slightly in comparison to those observed for the corresponding species in CO-matrixes, indicating weak interactions between the neighboring counterpart species formed after the bimolecular reaction of HOSO• with CO inside the rigid CO-matrix cages. For instance, the two characteristic IR bands of t-HOCO• for the $\nu$(O–H) and $\nu$(C=O) modes at 3456 and 1833 cm⁻¹ in CO-matrix shift to 3311.0 and 1831.7 cm⁻¹ due to complexation with SO (Supplementary Table 1). Concomitantly, the IR band of the counterpart SO at 1139.5 cm⁻¹ (CO-matrix) undergoes blue-shift to 1140.0 cm⁻¹, and the deformation mode $\delta$(COH)

exhibits a larger blue-shift of +12.8 cm⁻¹. The presence of the less stable conformer c-HOCO• is evidenced by the band at 1795.2 cm⁻¹, and it is also slightly red-shifted comparing to the band at 1797 cm⁻¹ for c-HOCO• in CO-matrix[18]. Conformational conversion of c-HOCO• to the lower-energy t-HOCO• happens upon subsequent irradiation at 532 nm. However, the previously reported[45] spontaneous transformation of c-HOCO• → t-HOCO• in N₂-matrix (4.5 K) via quantum mechanical tunneling was not observed in CO-matrix (16 K), which is consistent with the frequently observed environmental effects on the tunneling processes in low-temperature matrixes[48].

Consistent with the photodecomposition of HOSO• (→ H• + SO₂) in Ar-matrix, its photolysis in solid CO also causes H−O bond fragmentation followed by CO-trapping of the mobile hydrogen atoms to afford HCO•, which acts as a hydrogen donor through weak interaction with the counterpart SO₂ in the same CO-matrix cage by forming complex HCO•···SO₂. In this complex, the $\nu$(C-H) mode shifts to 2493.8 cm⁻¹ in comparison to the same mode at 2488 and 2483 cm⁻¹ for HCO• in CO- and Ar-matrixes[49]. The $\nu$(C = O) and $\delta$(COH) modes in HCO• and the two stretching modes of SO₂ display small red-shifts (Supplementary Table 2) in the complex. The changes of the two stretching modes of SO₂ in HCO•···SO₂ ($\Delta\nu_{asym}$ = −8.2 cm⁻¹; $\Delta\nu_{sym}$ = −0.3 cm⁻¹) are smaller than those observed in other SO₂-contaning complexes such as H₂O₂···SO₂ ($\Delta\nu_{asym}$ = −12.9 cm⁻¹; $\Delta\nu_{sym}$ = −2.1 cm⁻¹)[50]. Further combination of HCO• with mobile hydrogen atoms in the matrix during the laser photolysis affords H₂CO, and it is also perturbed by the neighboring molecule (SO₂) as evidenced by the appearance of two IR bands at 1736.7 and 1734.9 cm⁻¹ for the $\nu$(C = O) mode in H₂CO. Note, formation of HCO• and H₂CO has been previously observed in a solid CO/H₂ ice mixture at 8 K after irradiation with ultra-high vacuum UV (-160 nm) light[19].

The mechanism for the formation of HOS• during the photolysis of HOSO• in CO-matrix is intriguing. A plausible pathway is the HAT in

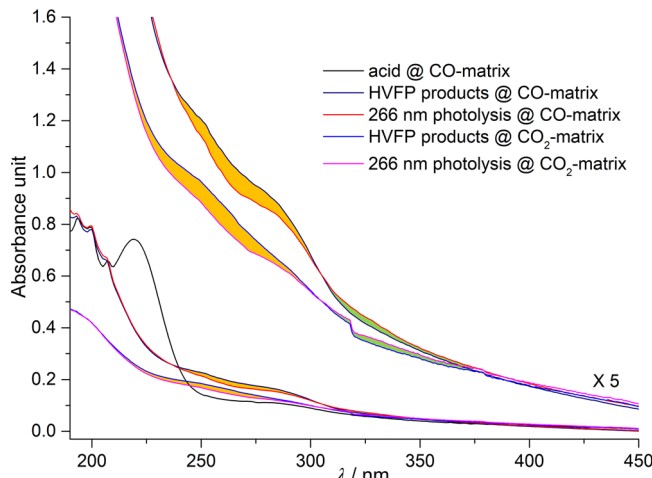

**Fig. 3 | Ultraviolet-visible (UV–vis) spectra of the acid precursor CHF₂S(O)OH and its high-vacuum flash pyrolysis (HVFP) products isolated in solid CO and CO₂ ices at 16 K.** The absorptions for HOSO• and its photolytic reaction products in the matrixes are rendered with orange and green colors, respectively.

HOCO•···SO ($\rightarrow$ CO$_2$···HOS•). In the CO$_2$···HOS• complex, the $\nu$(O–H) mode shifts by −154.3 cm$^{-1}$ in comparison with the same mode observed in HOS• in solid *para*-H$_2$[51], while the $\nu$(S–O) mode displays a blue-shift of +8.9 cm$^{-1}$ (Supplementary Table 3). Accordingly, the two IR bands for CO$_2$ at 2346.7 and 659.3 cm$^{-1}$ exhibit shoulders that can be assigned to the complex (Fig. 5). Particularly, the bending mode $\delta$(CO$_2$) in the complex appears as a broad band in the range of 665–650 cm$^{-1}$ due to removal of vibrational degeneracy upon complexation with HOS•. Similar splitting for the nondegenerate $\delta$(CO$_2$) modes has been observed in other CO$_2$-complexes such as HKrCCH···CO$_2$[52].

The spectroscopic identification of the photoproducts of HOSO• in CO-matrix is further supported by $^{18}$O-isotope labeling experiments. Using $^{18}$O-labeled sulfinic acid as the precursor, a 1:1:1:1 mixture of HOSO•, H$^{18}$OSO•, HOS$^{18}$O•, and H$^{18}$OS$^{18}$O• can be generated, and their distinction can be assured with the revolved IR bands at 1049.7, 1045.2, 1043.4, and 1039.5 cm$^{-1}$ for the $\delta$(SOH) mode in the four isotopologues, however, the remaining IR fundamental modes appear as doublets due to closeness of the isotopic shifts (Fig. 4d). The photochemistry of these isotopologues in CO-matrix (Fig. 4e) provides useful information for probing the reaction mechanism. The sole formation of HCO• (without HC$^{18}$O•) and a 1:2:1 mixture of SO$_2$, OS$^{18}$O, and S$^{18}$O$_2$ among the photoproducts confirms the route for the straightforward HAT from HOSO• to CO. The absence of the IR bands for HOC$^{18}$O• and H$^{18}$OC$^{18}$O• rules out the possibility for the formation of HOCO• via direct hydrogenation of CO$_2$, since the singly $^{18}$O-enriched CO$_2$ is present in the same matrix. In contrast, the association of the photolytically generated •OH/•$^{18}$OH with CO yields the experimentally observed HOCO• and H$^{18}$OCO• in 1:1 ratio.

In the IR difference spectrum for the $^{18}$O-labeling experiments (Fig. 4e), each band for the $\nu$(O–H) and $\nu$(S–O) modes in HOS• splits into doublet due to the additional presence of H$^{18}$OS•, the corresponding $^{16/18}$O-isotopic shifts −10.8 and −31.5 cm$^{-1}$ show good agreement with the calculated values −8 and −31 cm$^{-1}$, respectively. Assuming a bimolecular reaction of HOCO• and SO for the formation of HOS• and CO$_2$, the observation of more CO$_2$ than HOS• (based on the experimental and calculated IR band intensities) in the photochemistry indicates that there is an alternative pathway for producing CO$_2$. Given the generation of HO• and SO in the photolysis of HOSO•, the photochemistry of SO in CO ice was also studied. Co-condensation of gaseous SO in the presence of CO (ca. 1:1000) at 16 K also yields OSSO, which can be completely destroyed by UV-light irradiation at 365 nm (Supplementary Fig. 3). The formation of SO$_2$ and OCS

coincides with the photodissociation of OSSO to SO$_2$ and sulfur atom with subsequent CO-trapping reaction (S + CO $\rightarrow$ OCS). When changing the irradiation source to a 193 nm laser, depletion of SO occurs by forming CO$_2$ and OCS (Supplementary Fig. 3), and the mechanism can be reasonably explained that monomeric SO dissociates to sulfur and oxygen atoms followed by association reactions with CO. Therefore, formation of CO$_2$ and OCS in the photochemistry of HOSO• in solid CO is very likely caused by the photofragmentation of the initially generated SO. This mechanism is also consistent with the sole observation of none-isotopically labeled OCS and 1:1 mixture of CO$_2$/OC$^{18}$O in the $^{18}$O-lalebing experiments (Fig. 4e). Additionally, traces of HCO• form during laser irradiation of the impurity H$_2$O in solid CO.

The photolytic depletion of HOSO• in CO-matrix at 266 nm is extremely fast, and the depletion becomes much less efficient under subsequent UV-light irradiation at 365 nm (Supplementary Fig. 4 and 5). Prolonged 266 nm laser irradiation (36 min) leads to complete depletion of HOSO•. Further irradiation at 365 nm promotes reverse HAT from HOCO• to SO$_2$ by reforming HOSO• and CO$_2$. The photosensitivity of HOCO• towards the UV-light (365 nm) strongly indicates that the aforementioned weak absorptions in the UV-vis spectra in the range of 300 − 400 nm (Fig. 3) belongs to this carbonyl radical. It is also consistent with the observation in the IR spectra that HOCO• forms after the photolysis of HOSO• in CO-matrix (Fig. 4). The absorption for HCO• above 400 nm[53] was not observed in the UV-vis spectra due to low intensity.

The photo-induced HAT of HOSO• also occurs in CO$_2$-doped Ar-matrix at 16 K (Fig. 4c), yielding a new molecular complex HOCO•···SO$_2$ (Supplementary Table 1). The strongest band for the $\nu$(C = O) mode in the complex locates at 1818.9 cm$^{-1}$, and it is lower than the same mode in HOCO•···SO and HOCO• at 1831.7 and 1833 cm$^{-1}$, respectively. The very weak $\delta$(COH) mode undergoes small blue-shift to 1278.7 cm$^{-1}$ comparing to the frequencies of 1261 and 1265.8 cm$^{-1}$ for HOCO• in solid CO-matrix and in HOCO•···SO. In contrast, the two SO$_2$ stretching modes at 1350.9 and 1146.4 cm$^{-1}$ are red-shifted in comparison to free SO$_2$ at 1355.0 and 1152.2 cm$^{-1}$. Unlike the generation of two conformers of HOCO• in the photochemistry of HOSO• in CO-matrix (Fig. 4b), only one conformer was generated in the photolysis of HOSO• in CO$_2$-doped Ar-matrix (Supplementary Fig. 2). Additional formation of OCS among the photolysis products implies the trapping reaction of sulfur atoms (SO $\rightarrow$ S + O) by CO$_2$, however, the elusive intermediate OSCO[54] was not observed. Formation of OCS and HCO• was also observed during the 193 nm laser irradiation of a mixture of H$_2$O and SO$_2$ in CO-matrix (Supplementary Fig. 6).

## Calculated structures of molecule-radical complexes

Weakly bonded complexes consisting of simple radicals (e.g., HO• and HOO•) and small molecules (e.g., CO, CO$_2$, and H$_2$O) play important roles in gas-phase chemistry, as they may serve as key intermediates in clusters formation in atmospheric and astrochemical processes at the low-temperature surface of dust and ice grains[39]. Therefore, the structures and reactivity of these molecule-radical complexes have been the focus of comprehensive experimental and computational studies. Despite the importance of HOSO•, HOCO•, and HCO• in gas-phase chemistry has been increasingly recognized, their complexes remain scarcely investigated. Using the UCCSD(T)/aug-cc-pV(T + D)Z method, the structures, energies, and bonding properties for the new molecule-radical complexes involved in the photochemistry of HOSO• in CO and CO$_2$ ices were calculated (Fig. 6).

In sharp contrast to a favorable *cis*-planar structure for free HOSO•, the HOSO moiety in CO···HOSO• and CO$_2$···HOSO• are nonplanar. The global minimum of CO···HOSO• prefers OH···CO linkage and the hydrogen atom is dramatically tilted out of the OSO plane by 33.9°. The corresponding hydrogen bond length is 2.168 Å, which is shorter than the hydrogen bond in •OH···CO (2.341 Å)[55]. The binding energy ($D_e$ = 3.9 kcal mol$^{-1}$) is higher than those in •OH···CO ($D_e$ = 2.3 kcal mol$^{-1}$)[56]

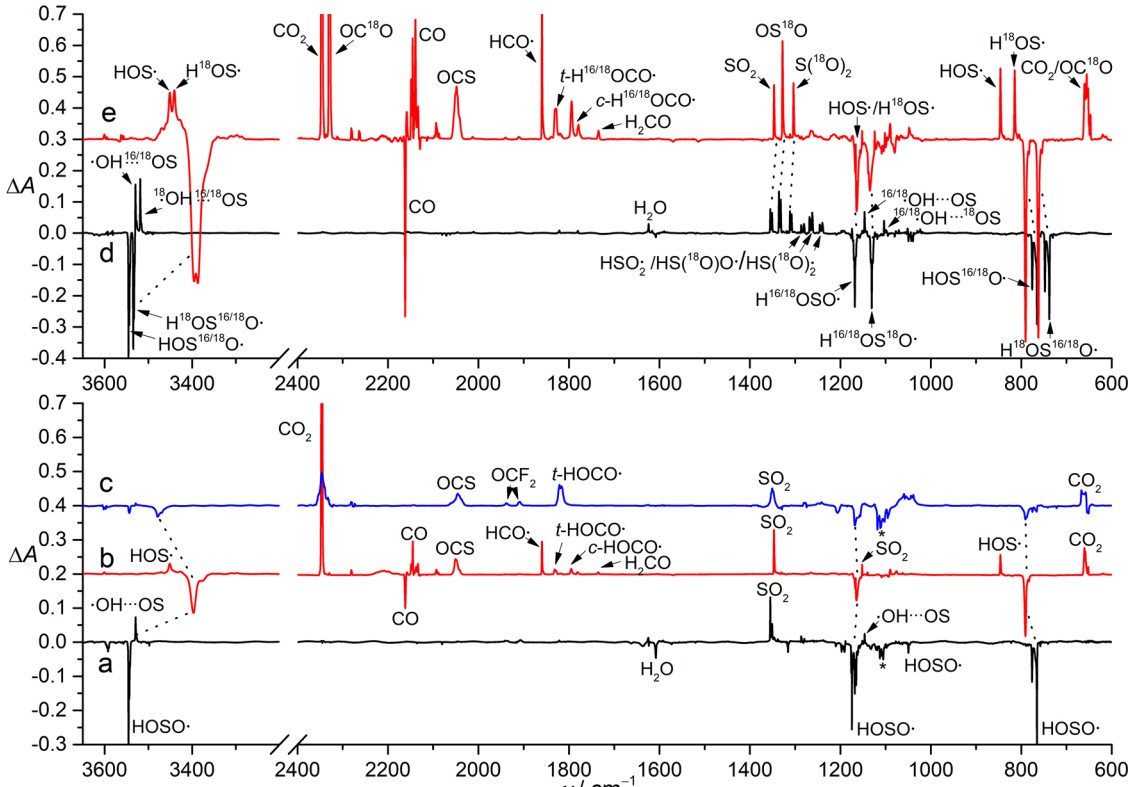

**Fig. 4 | Reactions of HOSO• in matrixes upon irradiation. a** Infrared (IR) difference spectrum reflecting the change of the Ar-matrix isolated high-vacuum flash pyrolysis (HVFP) products of $CHF_2S(O)OH$ upon irradiation at 266 nm (80 min, 10 K). **b** IR difference spectrum reflecting the change of the CO-matrix isolated HVFP products of $CHF_2S(O)OH$ upon irradiation at 266 nm (7 min, 16 K). **c** IR difference spectrum reflecting the change of the $CO_2$-doped Ar-matrix (50: 1000) isolated HVFP products of $CHF_2S(O)OH$ upon irradiation at 266 nm (30 min, 16 K). **d** IR difference spectrum reflecting the change of the Ar-matrix isolated HVFP products of $^{18}O$-labeled $CHF_2S(O)OH$ upon irradiation at 266 nm (32 min, 10 K). **e** IR difference spectrum reflecting the change of the CO-matrix isolated HVFP products of $^{18}O$-labeled $CHF_2S(O)OH$ upon irradiation at 266 nm (22 min, 16 K). The symbol "$^{16/18}O$" refers to a 1:1 mixture of the species containing $^{16}O$ and $^{18}O$.

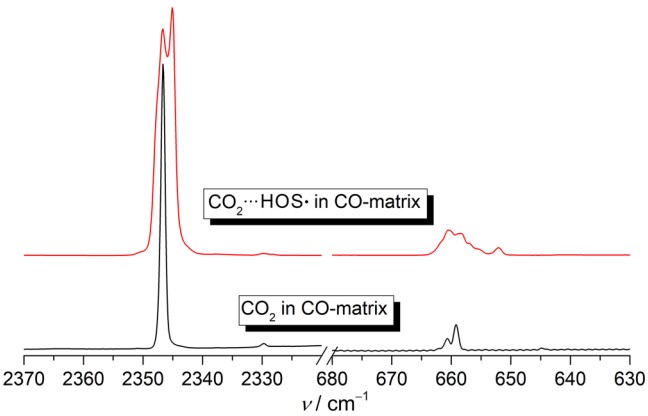

**Fig. 5 | Sections of Infrared (IR) spectra showing the two bands for the two vibrational modes of $CO_2$ in CO-matrix isolated $CO_2$ and $CO_2\cdots HOS•$.** The lower trace corresponds to the IR spectrum of $CO_2$ in CO-matrix at 16 K. The upper trace corresponds to the IR spectrum of $CO_2\cdots HOS•$ in CO-matrix at 16 K.

and •OH⋯OS ($D_e$ = 3.2 kcal mol$^{-1}$)[57]. The isomer bearing a OH⋯OC linkage between HOSO• and CO is less stable by ca. 5 kcal mol$^{-1}$. The hydrogen atom in $CO_2\cdots HOSO•$ is also tilted from the OSO plane by a dihedral angle of 30.5°, and the $CO_2$ molecule is slightly bent by 1.8°. The shorter hydrogen bond in $CO_2\cdots HOSO•$ (1.990 Å) than that in CO⋯HOSO• (2.168 Å) coincides with the larger stabilizing interaction ($D_e$ = 5.3 kcal mol$^{-1}$). The noncovalent interaction (NCI) analysis suggests strong attractive hydrogen bond interactions and simultaneous

weak repulsive interactions of the terminal S = O moiety with CO and $CO_2$ in these complexes (Fig. 6).

Unlike the distortion of the complexed *cis*-HOSO•, the HOCO moiety in the hydrogen-bonded HOCO•⋯SO and HOCO•⋯$SO_2$ complexes keeps a favorable *trans*-planar configuration. The shorter hydrogen bond in HOCO•⋯SO (1.893 Å) than that in the latter (1.909 Å) is also consistent with a higher stabilizing energy of 6.1 kcal mol$^{-1}$ (5.4 kcal mol$^{-1}$ in HOCO•⋯$CO_2$). The binding energy in HOCO•⋯SO is larger than that in the hydrogen-bonded $H_2O\cdots SO$ complex ($D_e$ = 3.1 kcal mol$^{-1}$)[58], while the latter remains yet experimentally unobserved. The weak interaction between HCO• and $SO_2$ in HCO•⋯$SO_2$ ($D_e$ = 3.5 kcal mol$^{-1}$) is facilitated by forming a five-membered ring through concerted contacts of hydrogen bond CH−OS (2.590 Å) and chalcogen bond CO−SO (2.978 Å). Similar five-membered ring structure has also been predicted for the detectable $HO_2$•⋯$SO_2$ complex ($D_e$ = 4.6 kcal mol$^{-1}$)[59]. The hydrogen bond interaction in the planar $CO_2\cdots HOS•$ complex ($D_e$ = 4.5 kcal mol$^{-1}$) is stronger than the intermolecular O⋯C contact in the T-shaped van der Waals complex $H_2O\cdots CO_2$ ($D_e$ = 2.8 kcal mol$^{-1}$)[60].

Energetically, HCO•⋯$SO_2$ and HOCO•⋯SO are higher than CO⋯HOSO• by 37.5 and 27.1 kcal mol$^{-1}$, respectively, whereas the secondary HAT in HOCO•⋯SO to form $CO_2\cdots HOS•$ is highly exothermic by releasing −51.5 kcal mol$^{-1}$. Therefore, the overall process for the oxidation of CO to $CO_2$ by reaction with HOSO• is exothermic by −14.0 kcal mol$^{-1}$, which is comparable with the energy (−20.0 kcal mol$^{-1}$) for the •OH radical promoted oxidation[14]. The HAT process in $CO_2\cdots HOSO•$ to form HOCO•⋯$SO_2$ is endothermic by 38.5 kcal mol$^{-1}$,

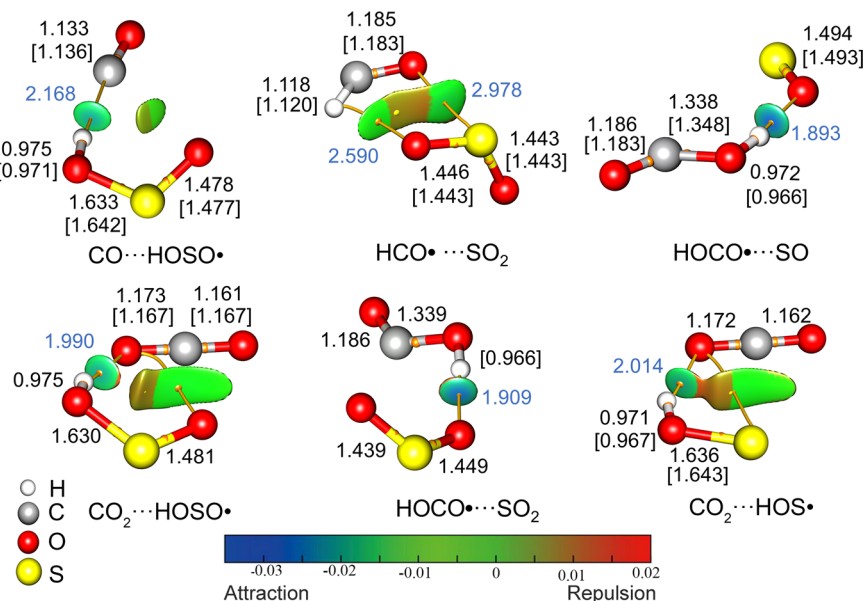

**Fig. 6 | The CCSD(T)/aug-cc-pV(T + D)Z calculated molecular structures and noncovalent interaction analysis for all the observed complexes.** The bond lengths (Å) for the monomers are given in square brackets, and the lengths for the intermolecular hydrogen bonds are shown in blue. The calculated gradient isosurfaces (s = 0.6 au) are colored on a blue-green-red scale according to values of sign($\lambda_2$)$\rho$, ranging from −0.035 to 0.02 au. Blue indicates strong attractive interactions, and red indicates strong repulsive interactions.

which is lower than the H−O bond dissociation energy in HOSO• (44.1 kcal mol⁻¹)[57].

## Implications in interstellar sulfur chemistry

Our experimental results demonstrate that organic radicals HCO• and HOCO• can be produced by UV-irradiation of HOSO• in astronomical CO and $CO_2$ ices via the hydrogen atom transfer (HAT) reactions of the initially formed molecular complexes at 16 K, and the acyl radicals also form stable molecular clusters with sulfur oxides through strong hydrogen bonding interactions. Considering the facile formation of HOSO• from the photoreactions of $SO_2$ with $H_2O$, the generation of the two important building blocks HCO• and HOCO• from the photoreactions mimics the chemical evolution network of the dehydrogenated sulfur-containing molecules SO and $SO_2$ in the outer layer of the CO-dominant interstellar icy grains in molecular clouds at a typical temperature of about 10 K. Alternatively, other radicals (e.g., •$CH_3$, •$CH_2OH$, and •CN) derived from the photoreactions of the interstellar carbon- or nitrogen-containing molecules (e.g., $CH_4$, $CH_3OH$, and HCN) in the cryogenic astronomical ices may also be present and react further with the acyl radicals through barrierless radical-radical association reactions to form more complex organic molecules (COMs). Additionally, the formation of OCS in the photochemistry of HOSO•, SO and $SO_2$ in CO and $CO_2$ ices may also contribute to understanding the interstellar sulfur chemistry, since OCS not only serves as a prebiotic activating agent for amino acid polymerization in forming peptides under mild conditions in aqueous solution[61–63], also it involves in the reduction of $CO_2$[64] and acts as a condensing agent in phosphate chemistry[65].

Clearly, the HAT processes of HOSO• may serve as the link connecting the chemistry of $SO_2$ and the chemistry of carbon oxides (CO and $CO_2$) in interstellar ices, although the ubiquity of HAT in chemistry, biology, and industry has been well recognized[66]. The uncovered chemical network for the formation of the astrochemically relevant organic radicals HCO• and HOCO• in the simple systems containing the primordial substrates (e.g., $H_2O$, SO, $SO_2$, CO, and $CO_2$) might aid in disclosing the intriguing mechanism for the chemical evolution of

biomolecules such as organic acids in dense molecular clouds, where barrierless radical-radical reactions at low temperatures (10–20 K) are assumed to happen spontaneously for the formation of COMs[67,68]. Furthermore, the spectroscopic identification and photochemistry of the new complexes consisting of the astrochemically relevant radicals HOSO•, HCO•, HOCO•, and HOS• will help in understanding the chemical composition and abundances in the interstellar medium, since molecular complexes are known to contribute to the formation of interstellar media and nucleation of aerosols in diverse planetary atmospheres, such as the CO-rich interstellar comet 2I/Borisov[69] and $SO_2$-rich atmosphere of Venus[70].

Aside from the role in ice-grain photochemistry, the planetary atmospheric chemistry of sulfur-containing species also attracts enormous interest due to importance in astrochemical reactions and planetary geology. Particularly, the photochemistry of $SO_2$ plays a fundamental role in the sulfur cycle in the Venusian atmosphere. In addition to the primary contribution from the volcanic eruptions[71], photolysis of $H_2SO_4$ vapor is another source of $SO_2$ as the planet is completely enshrouded by the acid droplets clouds. As demonstrated by previous field measurements[72] and modeling[34], photolysis of $H_2SO_4$ vapor yields $SO_3$ and $H_2O$, followed by further photodecomposition of $SO_3$ to $SO_2$ and SO. This simple photochemistry can enhance the abundances of $SO_2$ (66 ± 5 ppb) and SO (31 ± 4 ppb) in the cold (ca. −80 °C) mesosphere of Venus at altitudes of 85–100 km[73], where the CO abundances (ca. 30 ppm) vary with altitude partially due to reactions with sulfur compounds (e.g., $SO_2$, SO, and $S_2$)[74]. Hence, the disclosed photochemical reactions between sulfur oxides and carbon oxides in the presence of $H_2O$ (an average abundance of 30 ppm at 30–45 km altitude)[75] may affect the composition of Venusian atmosphere. Recently, Limaye et al.[76] proposed that the lower cloud layer of Venus (50–70 km) is an important target for study, since biorelevant organic acids might be generated through the iron-catalyzed metabolic redox reactions of the abundant $CO_2$, CO, $H_2O$, and $SO_2$ under favorable chemical and physical conditions. The observed photoreactions of sulfur-containing species with $CO_2$ and CO at low temperatures suggests that the hydrogenation of carbon oxides to the organic radicals HCO• and HOCO• for the formation of organic acids

might be facilitated by HAT reactions with the derived radicals HOSO•
and HSO•.

## Methods

### Sample preparation

Difluoromethylsulfinic acid ($CHF_2S(O)OH$) was synthesized by reaction
of hydrogen chloride (HCl) with sodium difluromethylsulfinate
($CHF_2S(O)ONa$)[44]. Specifically, freshly dried HCl (10 mmol) was con-
densed into a glass vessel containing solid $CHF_2S(O)ONa$ (0.14 g,
1 mmol) at −196 °C (liquid nitrogen bath), and the mixture was warmed
to −110 °C (cold ethanol bath) and kept for overnight reaction. Then,
the reaction mixture was slowly warmed to ca. −60 °C and the volatile
part was pumped (10 pa) through the vacuum line consisting of two
successive cold U-traps at −60 (cold ethanol bath) and −196 °C, and the
acid $CHF_2S(O)OH$ (ca. 50 mg, 0.5 mmol) was obtained in the first cold
trap. The $^{18}O$-enriched $CHF_2S(O)OH$ was synthesized through hydro-
lysis of $CHF_2S(O)Cl$ with water ($^{18}O$, 97%, Eurisotop), from which a 1:1:1:1
mixture of $CHF_2S(O)OH$, $CHF_2S(^{18}O)OH$, $CHF_2S(O)^{18}OH$, and
$CHF_2S(^{18}O)^{18}OH$ (Supplementary Fig. 7) was obtained according to the
IR spectrum of its decomposition product HOSO• and also the IR
spectrum of its reaction product ($SO_2$) with CO (Fig. 4e).

### Matrix-isolation spectroscopy

Matrix infrared (IR) spectra are recorded using an FT-IR spectrometer
(Bruker 70 V) in a reflectance mode with a transfer optic. A KBr beam
splitter and liquid-nitrogen-cooled mercury cadmium telluride (MCT)
detector are used in the mid-IR region (5000–450 $cm^{-1}$). For each
spectrum, 200 scans at a resolution of 0.5 $cm^{-1}$ are co-added. Matrix
ultraviolet-visible (UV–vis) spectra in the range of 190–800 nm are
recorded using a Perkin Elmer Lambda 850+ spectrometer with a
scanning speed of 1 nm $s^{-1}$.

For the preparation of the matrix, gaseous sample is mixed by
passing matrix gas (Ar, CO or $CO_2$) through a cold U-trap (−10 °C)
containing ca. 50 mg of the acid precursor ($CHF_2S(O)OH$). Then, the
mixture of acid vapor in matrix gas (a ratio of ca. 1:1000) is passed
through an aluminum oxide furnace (2.0 mm, i.d.:1.0 mm), which can
be heated over a length of ~25 mm by a tantalum wire (o.d. 0.4 mm,
resistance 0.4 Ω) and immediately deposited (2 mmol $h^{-1}$) in a high
vacuum (~$10^{-5}$ Pa) onto a gold-plated copper block matrix support
(10 K for Ar-matrix, 16 K for CO-matrix) for IR spectroscopy and onto a
$CaF_2$ window (16 K) for UV–vis spectroscopy using closed-cycle helium
cryostats (Sumitomo Heavy Industries, SRDK-408D2-F50H) inside the
vacuum chambers. For the preparation of the $CO_2$- or CO-doped Ar-
matrix, a premix of Ar with $CO_2$ or CO (a ratio of 1:20) was used as the
matrix gas. Temperatures at the second stage of the cold heads are
controlled and monitored using East Changing TC 290 (IR spectro-
scopy) and Lake-Shore 335 (UV–vis spectroscopy) digital cryogenic
temperature controller silicon diodes (DT-670). The voltage and cur-
rent use in the pyrolysis experiments are 7 V and 2.9 A, respectively.
Photolysis experiments were performed using the Nd$^{3+}$:YAG laser
(266 nm, MPL-F-266, 10 mW) and UV lamp (365 nm, 24 W).

### Quantum chemistry calculation

Molecular geometries of stationary points for the monomers and
complexes were first calculated at the B3LYP-GD3(BJ)/def2-TZVP[77]
level of theory with the Gaussian 16 software package[78]. The dispersion
correction using the D3 version of Grimme's dispersion with Becke-
Johnson damping, GD3(BJ)[79], is necessary to obtain reliable structures
for the hydrogen-bonded complexes. Then, the structures were fur-
ther optimized at the CCSD(T)/aug-cc-pV(T + D)Z[80] level of theory with
MOLPRO (ver. 2019.1) software[81]. The non-covalent interactions (NCIs)
analyses are carried out at the B3LYP-GD3(BJ)/def2-TZVP level of the-
ory with Bader's quantum atoms-in-molecules (QTAIM)[82] and John-
son's NCI analyses[83] on basis of the CCSD(T)/aug-cc-pV(T + D)Z level of
theory optimized structures.

## Data availability

All data to evaluate the conclusion in the paper are available in the
main text and/or the Supplementary Materials. The atomic coordi-
nates generated in this paper have been deposited in the ZENODO
database. (https://zenodo.org/record/7262494#.Y1zQn3ZBy3B).

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

## Acknowledgements

This work is supported by the National Natural Science Foundation of China (Grant 22025301 and Grant 22206027) and China Postdoctoral Science Foundation (Grant 2021M700802).

## Author contributions

X.Z. conceived project and designed the experiments. X.L. performed the synthesis and spectroscopic measurements. B.L., L.W., J.X and B.Z. also contributed to the spectroscopic measurements and analysis of the spectral data. T.T. carried out the quantum chemical calculations. X.Z., X.L. and J.S.F. discussed the results and drafted the manuscript. All authors commented on the manuscript.

## Competing interests

The authors declare no competing interests.
