## [Peer Review File · Nature Communications]

Unraveling Sulfur Chemistry in Interstellar Carbon Oxide IcesReviewer #1 (Remarks to the Author):

In this manuscript, the authors claim to show a link between interstellar COM formation and sulfur chemistry. To do this, they prepared HOSO in CO ice and identified the products after UV irradiation. The link, claimed by the authors is H atom transfer (HAT) from HOSO to CO, which results in the formation of the reactive HCO radical. On the one hand, the experiments are well planned, the assignments and the conclusions are well justified, the computations were carried out at a high level of theory, and the paper is very interesting. On the other hand, I do believe that for publication in Nature Communications, the manuscript has to be revised on some points.

Now the paper, especially the introduction is focusing on too many points: 1) Sulfur(oxide) chemistry in general in ISM, 2) HCO formation by HAT from HOSO, 3) structure of complexes,.....

I think, since the linked is the HAT transfer, the authors should focus on the role of HAT transfer from HOSO to CO. I think, it is more or less well-done in the Implications in Interstellar Sulfur Chemistry and in the Conclusion, but not in the other parts of the paper. I believe, that the authors could justify the importance of their findings if they can show that there are conditions/environments, where HCO formation could mostly be based on HAT from HOSO. For this, they have to show/discuss that the HOSO \rightarrow OSO + H bond rupture is very efficient, more efficient than alternative sources and/or there are astrophysical ice environments where the HOSO mixing ratio is expected to be high. Furthermore, from astrophysical relevance, broadband UV irradiation is more important than 266 nm irradiation. Also, I agree that the computation of the structures (and spectra) of the complexes is very important, but only those aspects should be discussed in the main manuscript which helps to understand the reaction mechanism. All the other aspects and discussion should be moved to the supporting information.

Minor points:

1) The authors write "According to the very recent theoretical study by Ruiz-López et al., HOSO is a very strong acid..." There is no direct link between acidity (HOSO \rightarrow OSO⁻ + H⁺) and HOSO \rightarrow OSO + H. They should only discuss the bond dissociation energy HOSO \rightarrow OSO + H, not to mention the acidity. (Or they should add a comment.)

2) "ultra-high vacuum (a'160 nm) light" should be "ultra-high vacuum UV (a'160 nm) light"

3) Which 18O labelled sulfonic acid was used? How/why do they get 1:1:1:1 mixture? (Should be explained briefly.)

4) "the observation of more CO₂ than HOS" In Figure 5 the A is displayed. To get molar ratios the absorptions should be normalized by the molar absorptions (infrared intensities). This is not shown, not discussed.

5) "The photo-induced hydrogen atom transfer of HOSO....(Fig 2C)." Is it Fig 3C? ("Hydrogen atom transfer" can be written as HAT.)

6) ":... (NCI) analysis suggest" should be "suggests"

7) "SO and SO₂ with hydrogen atoms arising from the decomposition of H₂ 49" " It is a bit misleading in ref. 49 H atoms were reacted by SO and SO₂. The authors are right that the original source of H atoms is H₂ molecules, but H atoms were generated in a series of photochemical reactions, which have no astrophysical relevance. The astrophysical conclusion of that paper is that H atoms react with SO and SO₂.

Reviewer #2 (Remarks to the Author):

The authors report on their studies on the formation of HCO and HOCO radicals from UV photochemistry of CO and CO₂ ices in the presence of the HOSO radical. These ices serve as analogues for modelling interstellar chemistry on small ice particles. HCO and HOCO radicals are important as nodes in radical reaction networks for forming higher order organic molecules in interstellar ice analogues, e.g., pyruvate and glyoxylate. Inorganic sulfur photochemistry is also important in the context of the atmospheric chemistry of planets and exoplanets, and so the HOSO radical could be important for forming prebiotic carbon feedstocks in the gas phase. The authors utilize HVFP of CHF₂S(O)OH to generate the HOSO radical, which they then condense back into the solid-phase for further photochemical investigations at low temperature by laser irradiation at 266 and 365 nm. Products were identified by IR and UV/Vis spectroscopy, aided and abetted by quantum mechanical calculations. The authors conclude that the HCO and HOCO radicals are obtained from hydrogen atom transfer of the HOSO radical under 266 nm irradiation in CO and CO₂ ices. The authors further suggest this chemistry could also be relevant to the SO₂-rich atmosphere of Venus.

Overall, this is an interesting and well-written paper, and the work will be of interest to prebiotic chemists, both gas-phase and potentially aqueous-phase chemists. The conclusions are well founded for the most part. The reviewer recommends publication after the following comments/criticisms have been addressed.

1) The authors use HVFP of CHF₂S(O)OH to generate HOSO radicals presumably as an experimental convenience for their photolysis studies. But, it would be insightful to know if the same HCO and HOCO radicals can be generated from photolysis of an astrochemically realistic icy mixture, perhaps one consisting of SO₂, CO and CO₂ ices. Could the authors please comment on the plausibility of this type of experiment, and perhaps even include some preliminary data?

2) While the experiments the authors report definitely seem relevant to the photochemistry of dense molecular clouds containing icy grains, the connection to atmospheric chemistry is more speculative. The authors dedicate a significant amount of discussion to potential gas-phase photochemistry, but the current manuscript contains no actual photochemical gas-phase data. The Reviewer thinks the manuscript would be much stronger if they can demonstrate, for example, gas-phase production of glyoxylic or pyruvic acid, by irradiation of, for example, sulfur oxide or carbon oxide gaseous mixtures. Can the authors please comment on whether or not they think such an experiment is feasible and if it can be included in the present manuscript?

3) Carbonyl sulfide is another interesting prebiotic molecule, as it has been demonstrated to be capable of serving as a prebiotic activating agent for amino acid polymerization. See the following reference for an example.

<https://www.science.org/doi/full/10.1126/science.1102722>

The authors might include a couple references to this point and a brief mention in the main text.

4) On page 13, the authors say "The photo-induced hydrogen atom transfer of HOSO also occurs in solid CO₂ at 16 K (Fig. 2C), yielding a new molecular complex HOCO...SO₂ (Table S1)." Do the authors mean Fig. 3C? Also, the authors should specifically mention how they prepared the CO₂ ice experiments in the SI. I assume "CO₂-doped Ar-matrix" has the same meaning as "solid CO₂", correct?

Reviewer #1

General comment: *In this manuscript, the authors claim to show a link between interstellar COM formation and sulfur chemistry.....the experiments are well planned, the assignments and the conclusions are well justified, the computations were carried out at a high level of theory, and the paper is very interesting. On the other hand, I do believe that for publication in Nature Communications, the manuscript has to be revised on some points.*

Now the paper, especially the introduction is focusing on too many points: 1) Sulfur(oxide) chemistry in general in ISM, 2) HCO formation by HAT from HOSO, 3) structure of complexes, I think, since the linked is the HAT transfer, the authors should focus on the role of HAT transfer from HOSO to CO. I think, it is more or less well-done in the Implications in Interstellar Sulfur Chemistry and in the Conclusion, but not in the other parts of the paper. I believe, that the authors could justify the importance of their findings if they can show that there are conditions/environments, where HCO formation could mostly be based on HAT from HOSO. For this, they have to show/discuss that the $\text{HOSO} \rightarrow \text{OSO} + \text{H}$ bond rupture is very efficient, more efficient than alternative sources and/or there are astrophysical ice environments where the HOSO mixing ratio is expected to be high. Furthermore, from astrophysical relevance, broadband UV irradiation is more important than 266 nm irradiation. Also, I agree that the computation of the structures (and spectra) of the complexes is very important, but only those aspects should be discussed in the main manuscript which helps to understand the reaction mechanism. All the other aspects and discussion should be moved to the supporting information.

Response: We greatly appreciate the reviewer's insightful comments. As suggested by the reviewer, the main focus of the present work is the discovery of a new pathway for the efficient formation of two astrochemically important acyl radicals $\text{HCO}\cdot$ and $\text{HOCO}\cdot$ through the hydrogen atom transfer (HAT) reactions of $\text{HOSO}\cdot$ in interstellar carbon oxide ices, and the observed HAT reactions may build the link connecting the inorganic chemistry of sulfur oxides and the organic chemistry of

carbon oxides in interstellar medium. As described in the Introduction, the two interstellar species $\text{HCO}\cdot$ and $\text{HOCO}\cdot$ are key building blocks for the formation of complex organic molecules (COMs) in interstellar medium. Previous laboratory studies have found that the two radicals can be generated through the hydrogenation of CO and CO_2 with hydrogen-containing molecules such as H_2 , H_2S , CH_4 , and H_2O , and vacuum ultraviolet (VUV) radiations were usually used due to the fact that these species absorb mainly below 200 nm. For an instance, formation of $\text{HOCO}\cdot$ in the photochemistry (130-170 nm, -70 min) of a solid $\text{H}_2\text{O}/\text{CO}$ mixture has been reported by Khriachtchev et al. (references 44 and 48). In sharp contrast, the formation of $\text{HOCO}\cdot$ from the UV-photolysis (266 nm, 7 min, Figure 3) of solid $\text{HOSO}\cdot/\text{CO}$ mixture becomes much more efficient, since $\text{HOSO}\cdot$ has an absorption in the range of 350-240 nm. This is also consistent with the Reviewer's comment that the H-O bond energy in $\text{HOSO}\cdot$ (44 kcal mol⁻¹) is significantly lower than that in H_2O (> 100 kcal mol⁻¹).

We also agree with the Reviewer that broadband UV irradiations are important in astrophysical chemistry. Indeed, we have also applied other light sources (193 laser, 365 nm LED, and high-pressure Hg-lamp) for the study of the photochemistry of $\text{HOSO}\cdot$ in CO ice. Formation of $\text{HCO}\cdot$ and $\text{HOCO}\cdot$ through HAT was also observed, this is also consistent with the broad UV-vis absorption band of $\text{HOSO}\cdot$ below 400 nm. However, the yields for the two acyl radicals and their complexes are much lower since the major absorption of $\text{HOSO}\cdot$ locates at around 270 nm (Figure 2). For clarity, the spectrum for the formation of $\text{HCO}\cdot$ during the 365 nm irradiation of $\text{HOSO}\cdot$ in CO ice has been added in the revised Supporting Information (Figure S5), whereas no IR bands for $\text{HOCO}\cdot$ were observed which is probably due to its photodissociation to $\cdot\text{H}$ and CO_2 . In the meantime, shorter-wavelength ArF excimer laser (193 nm) irradiation can deplete $\text{HCO}\cdot$ and $\text{HOCO}\cdot$ and their complexes. Therefore, the more selective 266 nm laser was applied to generate the intermediates (unstable complexes) in the HAT reactions of $\text{HOSO}\cdot$ in CO and CO_2 ices.

Fig. S5 | The photochemistry of HOSO• in CO-matrix. IR difference spectrum reflecting the change of the CO-matrix isolated HOSO• upon irradiation at 365 nm (7 min).

According to previous studies, dust grains in the interstellar medium typically consist of silicate or carbonaceous cores, whereas the interstellar ice mantles are predominantly composed of H₂O in the amorphous phase, combined with other molecules such as CO, CO₂, NH₃, CH₄, H₂CO, SO₂ and CH₃OH. Complex organic molecules (COMs) are supposed to be formed through H atom addition reactions in the CO or CO₂-rich ices via the intermediacy of reactive radicals such as HCO•, HOCO•, and •CH₂OH under various irradiations including energetic X-ray electrons, VUV light, and broadband irradiations, followed by barrierless recombination reactions under the irradiation conditions. In addition to the aforementioned generation of HCO• and HOCO• in the VUV photolysis of a H₂O/CO ice, formation of HCO• in the electron irradiation (5 keV) of a CH₃OH/CO mixture ice at 5 K was also observed (Kaiser et al. >. *Am. Chem. Soc.* **2021**, *143*, 14009-14018), and HCO• was proposed as a key building block for the prebiotic sugar formation. In order to compare with the efficient HAT reactions of HOSO• in solid CO ice, the photochemistry of a SO₂/H₂O/CO (1 : 10: 1000) mixture was also studied (Figure S6). As expected, only traces of HCO• without HOCO• were observed under the ArF

excimer laser irradiation conditions. Therefore, it is clear that hydroxidooxidosulfur radical ($\text{HOSO}\cdot$) acts as an important intermediate for formation of $\text{HCO}\cdot$ and $\text{HOCO}\cdot$ in solid CO ice.

Fig. S6 | The photochemistry of a $\text{SO}_2/\text{H}_2\text{O}/\text{CO}$ (1 : 10 : 1000) mixture. IR difference spectrum reflecting the change of the CO-matrix isolated H_2O and SO_2 upon irradiation at 193 nm (14 min).

As suggested by the Reviewer, Fig. 5 and Fig. 6 in the original manuscript have been move to the supporting information as Fig. S3 and Fig. S4, so as to put our focus on the HAT reactions of $\text{HOSO}\cdot$ to CO and CO_2 . Additionally, the introduction has also been shortened accordingly.

Minor Comment 1: *The authors write "According to the very recent theoretical study by Ruiz-Lopez et al., HOSO is a very strong acid..." There is no direct link between acidity ($\text{HOSO} \rightarrow \text{OSO}^- + \text{H}^+$) and $\text{HOSO} \rightarrow \text{OSO} + \text{H}$. They should only discuss the bond dissociation energy $\text{HOSO} \rightarrow \text{OSO} + \text{H}$, not to mention the acidity. (Or they should add a comment.)*

Response: The sentence, "According to the very recent theoretical study by Ruiz-Lopez et al., HOSO is a very strong acid with an estimated pKa value of -1.0 ", has been replaced by "In line with the low O-H bond dissociation energy (BED) of 44 kcal mol⁻¹ in HOSO•, this radical can behave as a hydrogen atom donor for HAT reactions".

Minor Comment 2): *"ultra-high vacuum (e160 nm) light"* should be *"ultra-high vacuum UV (e160 nm) light"*.

Response: Change has been made accordingly.

Minor Comment 3: *Which ¹⁸O labelled sulfonic acid was used? How/why do they get 1:1:1:1 mixture? (Should be explained briefly.)*

Response: The ¹⁸O-enriched CHF₂S(O)OH was synthesized through hydrolysis of CHF₂S(O)Cl with ¹⁸O-labeled water (97%, Eurisotop), from which an approximate 1:1:1:1 mixture of CHF₂S(O)OH, CHF₂S(¹⁸O)OH, CHF₂S(O)¹⁸OH, and CHF₂S(¹⁸O)¹⁸OH was obtained, based on the infrared spectrum of the decomposition product HOSO• in Ar-matrix at 10 K (Fig. 3E). The intensities for the characteristic IR bands of the photochemical products SO₂/OS¹⁸O/S¹⁸O₂ and HSO₂•/HS(¹⁸O)O•/HS¹⁸O₂• are 1:2:1, it is also consistent with a 1:1:1:1 mixture of ¹⁸O-enriched HOSO•, arising from the unimolecular decomposition of the ¹⁸O-enriched acid precursor. To make it clear, a scheme (Scheme S1) showing the generation of ¹⁸O-labeled HOSO• has been depicted in the Supporting Information.

Minor Comment 4: *"the observation of more CO₂ than HOS" In Figure 5 the A is displayed. To get molar ratios the absorptions should be normalized by the molar absorptions (infrared intensities). This is not shown, not discussed.*

Response: The ratio for the calculated IR intensities for the $\nu_{\text{asym}}(\text{CO}_2)$ and $\nu(\text{S-O})$ modes in HOS•-CO₂ complex is about 4 : 1. However, the experimentally observed ratio for the two bands is about 8 : 1, which means that there is an alternative pathway for the production of CO₂. As suggested by the reviewer, the corresponding

sentence has been changed to "Assuming a bimolecular reaction of HOCO• and SO for the formation of HOS• and CO₂, the observation of more CO₂ than HOS• (based on the experimental and calculated IR band intensities) in the photochemistry indicates that there is an alternative pathway for producing CO₂".

Minor Comment 5: "*The photo-induced hydrogen atom transfer of HOSO....(Fig 2C).*" Is it Fig 3C? ("*Hydrogen atom transfer*" can be written as HAT.)

Response: The typo has been changed.

Minor Comments 6: "*... (NCI) analysis suggest*" should be "*suggests*"

Response: The typo has been changed.

Minor Comments 7: "*SO and SO₂ with hydrogen atoms arising from the decomposition of H₂⁴⁹*", It is a bit misleading in ref. 49 H atoms were reacted by SO and SO₂. The authors are right that the original source of H atoms is H₂ molecules, but H atoms were generated in a series of photochemical reactions, which have no astrophysical relevance. The astrophysical conclusion of that paper is that H atoms react with SO and SO₂.

Response: To make it clear, the sentence "SO and SO₂ with hydrogen atoms arising from the decomposition of H₂" has been modified as "SO and SO₂ with hydrogen atoms".

Reviewer #2

General comment: *The authors report on their studies on the formation of HCO and HOCO radicals from UV photochemistry of CO and CO₂ ices in the presence of the HOSO radical. These ices serve as analogues for modelling interstellar chemistry on small ice particles. HCO and HOCO radicals are important as nodes in radical reaction networks for forming higher order organic molecules in*

interstellar ice analogues, e.g., pyruvate and glyoxylate. Inorganic sulfur photochemistry is also important in the context of the atmospheric chemistry of planets and exoplanets, and so the HOSO radical could be important for forming prebiotic carbon feedstocks in the gas phase. The authors utilize HVFP of CHF₂S(O)OH to generate the HOSO radical, which they then condense back into the solid-phase for further photochemical investigations at low temperature by laser irradiation at 266 and 365 nm. Products were identified by IR and UV/Vis spectroscopy, aided and abetted by quantum mechanical calculations. The authors conclude that the HCO and HOCO radicals are obtained from hydrogen atom transfer of the HOSO radical under 266 nm irradiation in CO and CO₂ ices. The authors further suggest this chemistry could also be relevant to the SO₂-rich atmosphere of Venus. Overall, this is an interesting and well-written paper, and the work will be of interest to prebiotic chemists, both gas-phase and potentially aqueous-phase chemists. The conclusions are well founded for the most part. The reviewer recommends publication after the following comments/criticisms have been addressed.

Response: We greatly appreciate the Reviewer's positive recommendations.

Comments 1: *The authors use HVFP of CHF₂S(O)OH to generate HOSO radicals presumably as an experimental convenience for their photolysis studies. But, it would be insightful to know if the same HCO and HOCO radicals can be generated from photolysis of an astrochemically realistic icy mixture, perhaps one consisting of SO₂, CO and CO₂ ices. Could the authors please comment on the plausibility of this type of experiment, and perhaps even include some preliminary data?*

Response: Thank you very much for your stimulating comment. The formation of HOSO from photolysis of a mixture of H₂S/H₂O in solid *para*-H₂ matrix and also in the gas phase through discharge of a SO₂/H₂ mixture has already been observed (references 21 and 22). As suggested by the Reviewer, we performed additional experiments on the photochemistry of mixtures of H₂O/SO₂/CO in different ratios. An shorter-wavelength ArF excimer laser at 193 nm was used for the photolysis,

since no noticeable changes occur to the mixture under the 266 nm laser irradiation conditions. The results for the typical experiment of a mixture of SO₂/H₂O/CO (1 : 10 : 1000) have been depicted in Fig. S5 in the Supporting Information. According to the IR difference spectrum, no IR bands for HOSO• and HOCO• but HCO• were observed. This is consistent with the previously observed photochemistry of HOSO• and HOCO• that both radical can decompose under irradiations at 193 nm. As suggested by the Reviewer, additional experiments for the photochemistry of a H₂O/SO₂/CO mixture with other light sources including VUV light and also electron beams are planned in our lab.

Fig. S6 | The photochemistry of a SO₂/H₂O/CO (1 : 10 : 1000) mixture. IR difference spectrum reflecting the change of the CO-matrix isolated H₂O and SO₂ upon irradiation at 193 nm (14 min).

Comments 2: *While the experiments the authors report definitely seem relevant to the photochemistry of dense molecular clouds containing icy grains, the connection to atmospheric chemistry is more speculative. The authors dedicate a significant amount of discussion to potential gas'phase photochemistry, but the current*

manuscript contains no actual photochemical gas-phase data. The reviewer thinks the manuscript would be much stronger if they can demonstrate, for example, gas-phase production of glyoxylic or pyruvic acid, by irradiation of, for example, sulfur oxide or carbon oxide gaseous mixtures. Can the authors please comment on whether or not they think such an experiment is feasible and if it can be included in the present manuscript?

Response: We fully agree with Reviewer that gas-phase reactions of HOSO• with CO and CO₂ should be of interest in the atmospheric chemistry, as HOSO• has already been found to be a key intermediate in the atmospheric SO₂ photochemistry at the air-water interface according to the very recent theoretical studies (see references 5 and 6). However, experimental studies about the gas-phase reactions of HOSO• with atmospherically relevant species are difficult to perform due to the lack of a suitable method for its gas-phase generation at "moderate" conditions and also the extremely high reactivity of the radical. The generation of HOSO• from high-temperature decomposition of the acid precursor permits the isolation and subsequent study on the photochemistry of this important species at low temperatures, including the photochemistry with CO and CO₂ in the present work. Further studies on the photolytic reactions of HOSO• with other interstellar species including a CO/CO₂ mixture, H₂O, and also organic molecules (e.g., CH₃CH₂OH and CH₃C(O)H) for possible production of biorelevant complex organic molecules such as glyoxylic acid and pyruvic acid through barrierless radical-radical association reactions of acyl radicals HCO•, HOCO•, and CH₃CO• in interstellar ices will be performed by using matrix-isolation spectroscopy. In order to detect the unstable reaction intermediates/products in the gas phase, a new chirped pulse Fourier transform millimeter wave (CPFTmmW) spectroscopy is under construction in our lab. It is noteworthy that barrierless radical-radical recombination reactions have been proposed as a key pathway for the chemical evolution of interstellar species within interstellar analog ices on ice-coated interstellar nanoparticles (see reference 71: Turner, A. M. & Kaiser, R. I. Exploiting photoionization reflectron time-of-flight mass spectrometry to explore molecular mass growth processes to

complex organic molecules in interstellar and Solar system ice analogs. *Acc. Chem. Res.* **53**, 2791-2805 (2020)).

Comments 3: *Carbonyl sulfide is another interesting prebiotic molecule, as it has been demonstrated to be capable of serving as a prebiotic activating agent for amino acid polymerization. See the following reference for an example. <https://www.science.org/doi/full/10.1126/science.1102722>. The authors might include a couple references to this point and a brief mention in the main text.*

Response: We thank the Reviewer for bringing our attention to the importance of OCS in the interstellar chemistry. As suggested by the Reviewer, a sentence has been added in the discussion section for the interstellar sulfur chemistry: "Additionally, the formation of OCS in the photochemistry of HOSO• in interstellar analogous ices of CO and CO₂ may also contribute to the interstellar sulfur chemistry, since OCS not only serves as a prebiotic activating agent for amino acid polymerization in forming peptides under mild conditions in aqueous solution,⁶⁴⁻⁶⁶ also it involves in the reduction of CO₂⁶⁷ and acts as a condensing agent in phosphate chemistry.⁶⁸". Moreover, a couple references have been added as 64-68:

64. Leman, L., Orgel, L. & Ghadiri, M. R. Carbonyl sulfide-mediated prebiotic formation of peptides. *Science* **306**, 283-286 (2004).
65. Frenkel-Pinter, M., Samanta, M., Ashkenasy, G. & Leman, L. J. Prebiotic peptides: Molecular hubs in the origin of life. *Chem. Rev.* **120**, 4707-4765 (2020).
66. Nair, N. N., Schreiner, E. & Marx, Dominik. Peptide synthesis in aqueous environments: the role of extreme conditions on amino acid activation. *Am. Chem. Soc.* **130**, 14148-14160 (2008).
67. Heinen, W. & Lauwers, A. M. Organic sulfur compounds resulting from the interaction of iron sulfide, hydrogen sulfide and carbon dioxide in an anaerobic aqueous environment. *Origins Life Evol. Biosphere* **2**, 131-150 (1996).

68. Biron, J. -P. & Pascal, R. Amino acid *N*-carboxyanhydrides: Activated peptide monomers behaving as phosphate-activating agents in aqueous solution. *J. Am. Chem. Soc.* **126**, 9198-9199 (2004).

Comments 4: *On page 13, the authors say "The photo-induced hydrogen atom transfer of HOSO• also occurs in solid CO₂ at 16 K (Fig. 2C), yielding a new molecular complex HOCO." SO₂ (Table S1)." Do the authors mean Fig. 3C? Also, the authors should specifically mention how they prepared the CO₂ ice experiments in the SI. I assume "CO₂-doped Ar-matrix" has the same meaning as "solid CO₂", correct?*

Response: The typo has been corrected. In addition to the experiments in pure CO and CO₂ solids, photochemistry of HOSO• in CO- and CO₂-doped Ar-matrixes was also performed. To make it clear, a sentence about the experimental details has been added in the supporting information: " For the preparation of the CO₂- or CO-doped Ar-matrix, a premix of Ar with CO₂ or CO (a ratio of 1:20) was used as the matrix gas."

Reviewer #1 (Remarks to the Author):

I am satisfied with the answers and the revision. The paper can be published in its present form.

Reviewer #2 (Remarks to the Author):

The reviewer would like to thank the authors for responding well to my first round of comments, but this reviewer agrees with the other reviewer that the introduction still lacks focus. Also, since the authors were unable to get the gas-phase data previously suggested (understandably so), the reviewer strongly feels the introduction and discussion sections need to have some major modifications.

The TOC graphic, abstract and title give the reader the impression that the context of the paper will be focused on the astrochemistry of interstellar space, i.e., ice grain photochemistry, and the experiments are certainly relevant to this context. The introduction, however, hardly even mentions this context. The first two paragraphs of the introduction are almost completely about atmospheric chemistry, which the experiments in the authors' manuscript do not directly relate to. The reviewer strongly suggests that the authors dedicate more space to giving context with respect to interstellar ices, and only briefly mention the potential relevance to planetary atmospheric chemistry.

The same criticism is applied to the discussion section – the relevance of the authors' results to interstellar ice-grain chemistry is virtually completely omitted, and instead the discussion is focused on Venusian atmospheric chemistry. The reviewer feels that the discussion section needs much more focus on interstellar ice-grain chemistry, and much less focus on pure speculation with respect to the organic chemistry of Venus's atmosphere. Figure 6 is especially problematic because the right-hand-side of the scheme is not supported by any data reported by the authors, especially the production of any carboxylic acid derivatives. Even if it were possible to synthesise relatively simple organic acids in this way, it is highly questionable that any other higher order organic molecule could be produced given the harsh conditions of the Venusian atmosphere, namely, the concentrated sulfuric acid which will rather quickly decompose organic molecules. Since the authors were not able to provide this sort of gas-phase data previously requested, understandably because these are very difficult experiments to perform, the reviewer strongly feels the authors need to tone down their speculations in this context, and focus more on the relevance to ice-grain astrochemistry.

In sum, the experiments of the authors are top-notch, but the introduction and discussion still need some work. After the changes suggested above are made, then publication in Nature Communications is recommended.

Thank you very much and also the reviewers' positive recommendations concerning our manuscript (Manuscript ID: NCOMMS-22-15206A). We also appreciate the second reviewer for the valuable suggestions on the introduction and discussion to improve the manuscript so that the focus is more relevant to the interstellar ice-grain chemistry. Revisions have been carefully made throughout the manuscript, and the corresponding changes have been highlighted in yellow in the Revised Manuscript. Moreover, our point-to-point responses to the Reviewer's comments are also given in below. We wish that our manuscript could be considered for publication in *Nature Communications* after the revision.

Sincerely yours,

Xiaoqing Zeng

Reviewer #1

Comment: **I am satisfied with the answers and the revision. The paper can be published in its present form.**

Author Response: **We greatly appreciate the reviewer's positive recommendation.**

Reviewer #2

Comment 1: *The reviewer would like to thank the authors for responding well to my first round of comments, but this reviewer agrees with the other reviewer that the introduction still lacks focus. Also, since the authors were unable to get the gas-phase data previously suggested (understandably so), the reviewer strongly feels the introduction and discussion sections need to have some major modifications. The TOC graphic, abstract and title give the reader the impression that the context of the paper will be focused on the astrochemistry of interstellar space, i.e., ice grain photochemistry, and the experiments are certainly relevant to this context. The introduction, however, hardly even mentions this context. The first two paragraphs of the introduction are almost completely about atmospheric chemistry, which the experiments in the authors' manuscript do not directly relate to. The reviewer strongly suggests that the authors dedicate more space to giving context with respect to interstellar ices, and only briefly mention the potential relevance to planetary atmospheric chemistry.*

Author Response: **We greatly appreciate the Reviewer for the very insightful comments. As enlightened by the Reviewer, we found that there are many recent studies about the (photo)chemistry in icy interstellar grains and analogues (e.g., solid CO and CO₂), which are formed as cold dense molecular clouds through fast condensation of gas-phase molecules at the surface of dust grain (mostly amorphous silicates) with an onion-like structure at the temperatures of about 10-20 K (for recent reviews, see: Materese, C. K. et al. Laboratory studies of astronomical ices: Reaction chemistry and spectroscopy. *Acc. Chem. Res.* 54, 280–290 (2021); Ferrari, B. C. et al. Role of suprathreshold chemistry on the evolution of carbon oxides and organics within interstellar and cometary ices. *Acc. Chem. Res.* 54 , 1067–1079 (2021); Turner, A. M. & Kaiser, R. I. Exploiting photoionization reflectron time-of-flight mass spectrometry to explore molecular mass growth processes to complex organic molecules in interstellar and Solar system ice analogs. *Acc. Chem. Res.* 53 , 2791–2805 (2020)). The inner**

layer of the grains mainly consists of hydrogenated ice (H_2O) with low concentrations of other H-containing species such as CH_3OH , NH_3 , and CH_4 . The outer layer is made up of dehydrogenated ices with dominant chemical compositions of CO , CO_2 , N_2 , O_2 , and SO_2 , and low concentrations of H_2O may also be present in the outer layer of the icy mantle. The icy mantle at the surface of cosmic dust grains are the most important carriers of prebiotic molecules, and the composition of the mantles are largely affected by the exchanges between solid ice and gas-phase and also the photochemistry promoted by cosmic irradiations, including UV and X-ray photons from nearby stars. Therefore, the study about the chemical composition of the icy grains and the involving reaction networks is crucial for understanding the chemical evolution of the molecular clouds and possibly providing new insight into the origin of life.

Carbon monoxide (CO) is the most abundant composition in the outer layer of icy grains in interstellar medium (ISM), and CO -abundant ices have also been found at the surface of many cold interstellar bodies, including comets, icy moons and planets in the outer solar system. So, the (photo)chemistry of CO through successive hydrogen atom addition reactions in this CO -rich layer may play a key role for the formation of diverse organic molecules, which are probably building blocks for the origin of life (Öberg, K. I. Photochemistry and astrochemistry: photochemical pathways to interstellar complex organic molecules. *Chem.Rev* 116 , 9631–9663 (2020); Chuang, K. -J. et al. Production of complex organic molecules: H-atom addition versus UV irradiation. *Mon. Not. R. Astron. Soc.* 467 , 2552–2565 (2017); Oba, Y. et al. An infrared measurement of chemical desorption from interstellar ice analogues. *Nat.Astron* 2 , 228–232 (2018)). Recent laboratory studies have demonstrated that biorelevant complex organic molecules (COMs) such as methylformate ($\text{CH}_3\text{OC(O)H}$), glycoaldehyde ($\text{HOCH}_2\text{C(O)H}$), glyoxylic acid (HC(O)C(O)OH), and pyruvic acid ($\text{CH}_3\text{C(O)C(O)OH}$) can be generated in astronomical CO ices (for examples, see: Eckhardt, A. et al. Formation of glyoxylic acid in interstellar ices: a key entry point for prebiotic chemistry. *Angew. Chem. Int.Ed* 58 , 5663–5667 (2019); Kleimeier, N. F. et al. Interstellar formation of biorelevant pyruvic acid ($\text{CH}_3\text{COCO}(\text{O})\text{H}$). *Chem* 6 , 3385–3395 (2020); Lim, R .W. J. & Fahrenbach, A. C. Radicals in prebiotic chemistry. *Pure Appl.Chem* 92 , 1971–1986 (2020)). Importantly, highly reactive radicals such as formyl radical ($\text{HCO}\bullet$) and hydroxycarbonyl radical ($\text{HOCO}\bullet$) generated through hydrogen atom addition to CO and CO_2 under UV-light irradiation conditions are found to be key intermediates for the formation of

these COMs through barrierless radical-radical association reactions via thermally initiated diffusion of the radicals in the cryogenic CO-ices at temperatures of about 10 K. More recently, the radical recombination reactions of HCO• during the phase transition of interstellar CO ice at a typical dense cloud temperature of 10 K has been also disclosed (He, J. et al. Radical recombination during the phase transition of interstellar CO ice. *Astrophys J* 931, L1–L6 (2022); He, J. et al. Phase transition of interstellar CO ice. *Astrophys. J.* 915, L23–L28 (2021)).

In sharp contrast to the extensively explored mechanisms for the formation of COMs through the photoreactions of H-containing species (e.g., CH₃OH, NH₃, and CH₄) via the intermediacy of organic radicals such as HCO•, HOCO•, and CH₃O• in astronomical CO and CO₂ ices, the astrochemistry of dehydrogenated molecules SO and SO₂ via the intermediacy of sulfur-containing radicals HSO• and HOSO• in astronomical CO and CO₂ ices receives much less attention (for examples, see: Mifsud, D. V. et al. Sulfur ice astrochemistry: A review of laboratory studies. *Space Sci. Rev.* 217, 14–47 (2021); Laas, J. C. & Caselli, P. Modelling sulfur depletion in interstellar clouds. *Astron. Astrophys.* 624, A108–A124 (2019)). However, sulfur is considered to be one of the six fundamental elements (along with H, C, O, N, and P) to the life, and sulfur is found in a wide variety of biomolecules including amino acids, nucleic acids, sugars, and vitamins. Recently, it has been shown that the hydrogenation of SO and SO₂ in solid H₂ can yield HSO• and HOSO• (Góbi, S., Csonka, I. P., Bazsó, G. & Tarczay, G. Successive hydrogenation of SO and SO₂ in solid *para*-H₂: formation of elusive small oxoacids of sulfur. *ACS Earth Space Chem* 5, 1180–1195 (2021)), and the latter has also been found to be efficiently formed from either the reaction of the photoexcited SO₂ at the surface of water (Martins-Costa, M. T. C. et al. Photochemistry of SO₂ at air-water interface: A source of OH and HOSO radicals. *J. Am. Chem. Soc.* 140, 12341–12344 (2018); Ruiz-López, M. F. et al. A new mechanism of acid rain generation of HOSO at the air-water interface. *J. Am. Chem. Soc.* 141, 16564–16568 (2019)) or the radical association of SO with •OH (Chen, C. et al. Capture of the sulfur monoxide–hydroxyl radical complex. *J. Am. Chem. Soc.* 142, 2175–2179 (2020)). More recently, the sulfur astrochemistry involving SO and SO₂ in the planetary atmospheres has also been intensively explored (Pinto, J. P. et al. Sulfur monoxide dimer chemistry as a possible source of polysulfur in the upper atmosphere of Venus. *Nat. Commun* 12, 175 (2021); Francés-Monerris, A. et al. Photochemical and thermochemical pathways to S₂ and polysulfur formation in the atmosphere

of Venus. *Nat. Commun* 13 , 4425 (2022)). It is noteworthy that the two sulfur oxides SO₂ and SO have also been found to be very abundant in molecular clouds (Rydbeck, O. E. H. et al. Observations of SO in dark and molecular clouds. *Astrophys. J* 235 , L171–L175 (1980); Irvine, W. M. et al. Observations of SO₂ and HCS⁺ in cold molecular clouds. *Astron Astrophys* 127 , L171–L175 (1983); Cernicharo, J. et al. Collisional excitation of sulfur dioxide in cold molecular clouds. *Astron. Astrophys* 127 , A103 (2011); Vidal, T. H. G. et al. On the reservoir of sulphur in dark clouds: chemistry and elemental abundance reconciled. *Mon. Not. R. Astron. Soc.* 469 , 435–447 (2017)). Considering the above mentioned importance of astrochemistry of carbon oxides (CO and CO₂) and sulfur oxides (SO and SO₂), herein, we report a first experimental study about the photochemistry of sulfur oxides in solid carbon oxides, in which the radical species HOSO• not only forms molecular complexes with carbon oxides but also acts as an hydrogenation agent for carbon oxides by efficient generation of the two organic radicals HCO• and HOCO•, which are key building blocks for the formation of COMs in icy interstellar grains.

Based on the Reviewer's suggestions and also the aforementioned background information, the first two paragraphs about the atmospheric chemistry of SO₂ and HOSO• in the introduction of the manuscript has been strongly shortened, while the ice grain photochemistry of H-containing species in interstellar CO and CO₂ ices involving the formation of COMs through the reactions of HCO• and HOCO• radicals has been included mostly at the end of the first paragraph and also in the beginning of the second paragraph: "Aside from the importance in planetary atmospheric chemistry, the two sulfur oxides SO₂ and SO have also been found to be very abundant in molecular clouds,¹³⁻¹⁶ which are mainly formed through condensation of gas-phase molecules at the surface of dust grains (mostly amorphous silicates) with an onion-like structure at the temperatures of about 10–20 K.^{17,18} The inner layer of the grains mainly consists of hydrogenated ice (H₂O) with low concentrations of other H-containing species such as CH₃OH, NH₃, and CH₄. The outer layer is made up of dehydrogenated ices with dominant compositions of CO, CO₂, N₂, O₂, and SO₂, and low concentrations of H₂O may also be present in the outer layer of the icy mantle. The icy mantle at the surface of cosmic dust grains are the most important carriers of prebiotic molecules, and the composition of the mantles are largely affected by the exchanges between solid ice and gas-phase and also in the photochemistry

promoted by cosmic irradiations, including UV and X-ray photons from nearby stars. Therefore, the study about the chemical composition of the icy grains and the complex reaction networks is crucial for understanding the evolution of the molecular clouds.¹⁹⁻²¹

Carbon monoxide (CO) is the most abundant composition in the outer layer of icy grains in interstellar medium (ISM), and CO-abundant ices have also been found at the surface of many cold interstellar bodies, including comets, icy moons and planets in the outer solar system. Therefore, the chemistry of CO through successive hydrogen atom addition reactions in the CO-rich outer layer may play a key role for the formation of organic molecules, which are probably building blocks for the origin of life.²²⁻²⁴ The beginning of the third paragraph has also been modified by addition of "In sharp contrast to the extensively explored mechanisms for the formation of COMs through the photoreactions of H-containing species (e.g., CH₃OH, NH₃, and CH₄) via the intermediacy of organic radicals such as HCO•, HOCO•, and CH₃O• in interstellar icy grain mantles, the ice-grain chemistry of the typical dehydrogenated molecules such as SO, SO₂, and the derived sulfur-containing radicals HSO• and HOSO• in astronomical CO and CO₂ ices remains barely investigated.". For consistency, the references have also been changed accordingly.

Comment 2: *The same criticism is applied to the discussion section—the relevance of the authors' results to interstellar ice-grain chemistry is virtually completely omitted, and instead the discussion is focused on Venusian atmospheric chemistry. The reviewer feels that the discussion section needs much more focus on interstellar ice-grain chemistry, and much less focus on pure speculation with respect to the organic chemistry of Venus's atmosphere. Figure 6 is especially problematic because the right-hand-side of the scheme is not supported by any data reported by the authors, especially the production of any carboxylic acid derivatives. Even if it were possible to synthesise relatively simple organic acids in this way, it is highly questionable that any other higher order organic molecule could be produced given the harsh conditions of the Venusian atmosphere, namely, the concentrated sulfuric acid which will rather quickly decompose organic molecules. Since the authors were not able to provide this sort of gas-phase data previously requested, understandably because these are very difficult experiments to perform, the reviewer strongly feels the authors need to tone down their speculations in this context, and focus more*

on the relevance to ice-grain astrochemistry.

Author Response: We completely agree with the Reviewer's suggestion that the interstellar ice-grain chemistry should be discussed, based on the disclosed photoreactions of sulfur oxides in astronomical CO and CO₂ ices. As for the discussion about the Venusian atmospheric chemistry, it mainly bases on the observed formation of the two reactive acyl radicals (HCO• and HOCO•) from the photoreactions as facilitated by hydrogen atom transfer (HAT) with the derived radicals HOSO• and HSO•, since both radicals are known as very important building blocks for the formation of complex organic molecules (organic compounds with six or more atoms as usually defined by astronomers) according to the recent studies in astronomical ices (for examples, see: Eckhardt, A. et al. Formation of glyoxylic acid in interstellar ices: a key entry point for prebiotic chemistry. *Angew. Chem. Int Ed* **58** , 5663–5667 (2019); Kleimeier, N. F. et al. Interstellar formation of biorelevant pyruvic acid (CH₃COCOOH). *Chem* **6** , 3385–3395 (2020); He, J. et al. Radical recombination during the phase transition of interstellar CO ice. *Astrophys J* **931** , L1–L6 (2022)). On the other hand, Limaye et al. have proposed recently that the lower cloud layer of Venus (50–70 km) is an important target for study, since biorelevant organic acids might be generated through the iron-catalyzed metabolic redox reactions of the abundant gaseous molecules CO₂, CO, H₂O, and SO₂ and probably can also survive under favorable chemical and physical conditions (please see the adapted figure in below from the corresponding reference: Limaye, S. S. et al. Venus' spectral signatures and the potential for life in the clouds. *Astrobiology* **18** , 1181–1198 (2018)). For clarity, the original figure showing the proposed pathways for the formation of organic acids (H_xC_yO_z) in this reference is shown below. According to our observation, the formation of the two major building blocks HCO• and HOCO• for the formation of organic molecules might occur alternatively through the hydrogen atom transfer reactions of carbon oxides by HOSO• and HSO•, which are key intermediates in the gas phase photochemistry of SO₂ and SO in the presence of H₂O. The generation of SO₂, SO, and H₂O during the photochemistry of sulfuric acid in the Venusian atmosphere has also been reported by Zhang, X. et al. (please see the adapted figure in below from the corresponding reference: Zhang, X. et al. Photolysis of sulphuric acid as the source of sulphur oxides in the mesosphere of Venus. *Nat. Geosci.* **3** , 834–837 (2010)). However, as pointed out by the Reviewer, at the moment we are not able to perform the corresponding gas-phase photoreactions for the

SO₂/H₂O/CO system. Nevertheless, we are now trying to build a new matrix-isolation machine by coupling with mass spectrometry and temperature-programmed desorption, so that we will be able to better understand the reactions of the photolytically generated radicals during the warm-up of the interstellar ices.

Redacted

Figure adapted from reference: Limaye, S. S. et al. Venus' spectral signatures and the potential for life in the clouds. *Astrobiology* **18** 1181-1198 (2018).

Redacted

Figure adapted from reference: Zhang, X. et al. Photolysis of sulphuric acid as the source of sulphur oxides in the mesosphere of Venus. *Nat. Geosci.* 3 (2010).

As kindly suggested by the Reviewer, Figure 6 was deleted, and the discussions about the implications in the chemistry of Venusian atmosphere were also shortened. In the meantime, another paragraph about the relevance of the experimental results to ice-grain astrochemistry has been added as "It is also clear that organic radicals FICO• and HOCO• are produced by UV-irradiation of HOS00 in astronomical CO and CO₂ ices via the hydrogen atom transfer (HAT) reactions of the initially formed molecular complexes at 10 K, and the acyl radicals also form stable molecular clusters with sulfur oxides through strong hydrogen bonding interactions. The facile generation of the two important building blocks NCO* and HOCO• from the photoreactions mimics the chemical evolution network of the dehydrogenated sulfur-containing molecules SO and SO₂ in the outer layer of the CO-dominant interstellar icy grains in molecular

clouds at a typical temperature of about 10 K. Alternatively, other radicals (e.g., $\bullet\text{CH}_3$, $\bullet\text{CH}_2\text{OH}$, and $\bullet\text{CN}$) derived from the photoreactions of the interstellar carbon- or nitrogen-containing molecules (e.g., CH_4 , CH_3OH , and HCN) in the cryogenic astronomical ices may also be present and react further with the acyl radicals through barrierless radical-radical association reactions by forming more complex organic molecules. Additionally, the formation of OCS in the photochemistry of SO and SO_2 in CO and CO_2 ices may also contribute to understanding the interstellar sulfur chemistry, since OCS not only serves as a prebiotic activating agent for amino acid polymerization in forming peptides under mild conditions in aqueous solution,⁶⁴⁻⁶⁶ also it involves in the reduction of CO_2 ⁶⁷ and acts as a condensing agent in phosphate chemistry.⁶⁸”

Comment 3: *In sum, the experiments of the authors are top-notch, but the introduction and discussion still need some work. After the changes suggested above are made, then publication in Nature Communications is recommended.*

Author Response: We greatly appreciate the Reviewer’s positive recommendation. We strongly agree with the Reviewer that laboratory studies about the (photo)chemistry in the analogous ice-grain mantles is of vital importance in astrochemistry. The recently disclosed formation of complex organic molecules and molecular clusters through the spontaneous recombination reactions of the reactive radicals (induced by UV light, cosmic rays, and energetic ion bombardment) in interstellar ices at a typical dense cloud temperature of 10 K in the lab helps in probing the composition, distribution, abundances, and also reaction networks of the molecules in the interstellar medium. In sharp contrast to the extensively explored astronomical reactions of the C-, O-, and N-containing radicals (e.g., $\text{HCO}\bullet$, $\text{HOCO}\bullet$, $\bullet\text{CH}_2\text{OH}$, $\bullet\text{CN}$, $\bullet\text{CCN}$, and $\bullet\text{CH}_2\text{CN}$), little is known about the astrochemistry for the biorelevant S- and P-containing radicals, whose existence has already been proposed in the astrochemistry of the interstellar molecules such as SO, SO_2 , H_2S , and PH_3 . We will continue working on the interstellar ice-grain chemistry through studies about the reactions of the radicals generated in the photochemistry of these interstellar molecules by combining matrix-isolation spectroscopy with mass spectrometry (under construction in our lab) and quantum chemical calculations.